# High Numbers of CD163-Positive Macrophages in the Fibrotic Region of Exuberant Granulation Tissue in Horses

**DOI:** 10.3390/ani11092728

**Published:** 2021-09-18

**Authors:** Charis Du Cheyne, Ann Martens, Ward De Spiegelaere

**Affiliations:** 1Department of Morphology, Ghent University, 9820 Merelbeke, Belgium; charis.ducheyne@ugent.be; 2Department of Surgery and Anaesthesiology of Domestic Animals, Ghent University, 9820 Merelbeke, Belgium; ann.martens@ugent.be

**Keywords:** exuberant granulation tissue (EGT), horse wound healing, proud flesh, pro-inflammatory macrophages, anti-inflammatory macrophages, MAC387, CD163

## Abstract

**Simple Summary:**

Horses are prone to develop a wound healing disorder on their limbs called exuberant granulation tissue (EGT). The exact mechanism for the formation of this tissue remains unknown but the inflammatory response is supposed to be an important contributing factor. In this article, we investigated this inflammatory response in both EGT wounds as well as in control horse wounds. In biopsies, we detected two types of immune cells: (1) immune cells involved in early inflammation (MAC387+ cells) and (2) immune cells important in the later phases of inflammation (CD163+ cells). We detected a higher number of immune cells in EGT wounds compared with the control wounds of 19 days old. This suggests that EGT wounds may not be able to proceed through further phases in the wound healing process or that the inflammation phase is prolonged.

**Abstract:**

Exuberant granulation tissue (EGT) is a frequently encountered complication during second intention healing in equine distal limb wounds. Although it is still unknown what exactly triggers the formation of this tissue, previous research has revealed a persistent inflammatory response in these wounds. In this preliminary study we examined this inflammatory response in EGT-developing wounds as well as in experimental induced wounds. Immunohistological stainings were performed to detect primary inflammatory immune cells (MAC387 staining) as well as pro-resolution immune cells (CD163 staining). Our results show a significantly higher amount of MAC387+ and CD163+ cells in the fibrotic regions of EGT compared with the 19-day-old experimental wounds. This persistent high amount of fibrosis-promoting CD163+ cells in EGT suggests that the wound healing processes in EGT-developing wounds are arrested at the level of the proliferation phase.

## 1. Introduction

Wound healing is an essential process for the health of animals but can also be associated with severe complications. Veterinarians are frequently confronted with horses that have serious wound healing issues at the level of the distal limbs. Wounds at these locations are often accompanied by massive tissue loss and typically need to heal by second intention, i.e., through the initial formation of granulation tissue that fills the wound bed and the subsequent re-epithelialization and wound contraction [1,2]. Granulation tissue is composed of collagen, elastin, proteoglycans and hyaluronic acid. Furthermore, it is characterized by the ingrowth of new blood vessels and the presence of fibroblasts, keratinocytes and inflammatory cells [3]. In horse distal limb wounds, it often occurs that this granulation tissue keeps proliferating beyond the margins of the wound bed preventing efficient epithelialization [4]. This tissue is called exuberant granulation tissue (EGT) or proud flesh. 

The exact physiopathology of EGT remains unclear but comparative studies have already revealed that differences in the immune response have a big impact on the formation of EGT [2,5]. In contrast to horses, ponies seldom develop EGT. Studies that compared the healing process of distal limb wounds of ponies with horses describe a short but fierce inflammatory response in ponies whereas horse wounds are characterized by a lower but persistent inflammatory response [5]. Moreover, body location seems to influence the development of EGT as EGT formation is typically seen on distal limbs [6,7]. Again, this is probably related to a different immune response. In a study conducted by Lepault et al. (2005), the inflammatory response in well-healing equine thoracic wounds was quick and resolved within three weeks whereas in limb wounds the inflammatory response was not resolved by six weeks [2]. In equine distal limb wounds, the number of inflammatory cells peaks approximately two weeks after injury but, despite a slight decrease, they are still markedly present six weeks after injury [2,5]. This chronic inflammation is believed to be one of the major factors stimulating the formation and maintenance of EGT [8]. Other factors that can be associated with EGT formation are differences in collagen metabolism and angiogenesis, improper bandaging, wound contamination, foreign bodies and iatrogenic factors [9].

In physiological healing wounds, the inflammatory response starts when immune cells such as neutrophils and monocytes are recruited to the wound bed. Monocytes can differentiate into macrophages. These highly plastic cells can change their phenotype and function throughout the wound healing process according to the stimuli present in the micro-environment [10]. In the early phases of wound healing, macrophages become activated to pro-inflammatory M1-type macrophages, which clear the wound bed from debris and pathogens by phagocytosis and produce pro-inflammatory cytokines and reactive oxygen species [10,11,12]. These pro-inflammatory immune cells are gradually replaced by pro-resolution M2-type macrophages, which resolve the initial inflammation and orchestrate the proliferation phase of wound healing [13,14]. M2 macrophages are involved in the formation of granulation tissue, re-epithelialization and the restoration of the blood vessel network [12,15,16,17]. In EGT, the inflammatory response is disrupted and characterized by a weak but persistent inflammation causing an excessive proliferation of the granulation tissue. 

In this article, different immunohistological and histological stainings were performed to gain more insights in the inflammatory response occurring in EGT.

## 2. Materials and Methods

### 2.1. Horses

Two types of wound healing samples were used, i.e., samples of physiological second intention healing experimental equine wounds as the control and exuberant granulation tissue from equine patients. The samples of physiological healing wounds were derived from a previous study (a non-contaminated control treatment) [18]. The original experiment was approved by the local ethical committee on 3 February 2014 (approval number 2014/183) [18,19]. Five warmblood horses (three geldings and two mares) ranging from 7 to 15 years of age were included in this study. Their body weight varied between 535 and 648 kg and both their metacarpi were free of scars. 

Exuberant granulation tissue was collected from patients presented at the Faculty of Veterinary Medicine of Ghent University. To be included in this study, it was important that EGT material could be obtained as fresh as possible. Apart for one exception, the wounds were at least two weeks old. EGT included in this study was obtained from twelve different warmblood horses (aged between 2 months and 14 years; 6 mares and 6 stallions). The weight was recorded for most horses and varied from 340 to 635 kg. 

### 2.2. Wounds

#### 2.2.1. Experimental Wounds

Circular wounds of 3.5 cm in diameter were created on the dorsomedial aspect of both metacarpi during an aseptic surgery. Both the skin and subcutis were removed using a scalpel. In the center of each wound, the periosteum was also removed over a circular area of 2 cm diameter with the aid of a sterile template. The underlying bone was curetted 15 times each in both a dorsopalmar and proximodistal direction. The wounds were first covered with a sterile non-adherent absorbent dressing (ZorbopadTM, Millpledge veterinary, Clarborough, United Kingdom), which was fixed using an elastic retention dressing (glatt LuxTM, Mai med, Neuenkirchen, Germany). The next layer consisted of a standard cotton wool limb bandage and the top layer was an elastic bandage (IdealflexTM, Hartmann, Nijmegen, The Netherlands). 

All the samples used in the current study originated from the non-contaminated control group. The day after surgery the bandages were removed and one limb was assigned to the treatment procedure and the other to the reference treatment. All the samples used in the current study originated from wounds cured with the reference treatment. For this option, the wounds were covered with calcium alginate dressings (cut into a square of 10 cm by 10 cm, Kendall, Medtronic, Minneapolis, MN, USA) soaked with a 20 mL sterile saline solution (0.9% NaCl). This primary dressing was molded to the wound to minimize overlap with the surrounding skin and covered with an elastic retention dressing (glatt LuxTM, Mai med, Neuenkirchen, Germany). A standard limb bandage was then applied, as described earlier. The primary wound dressing was calcium alginate for 9 days with bandage changes every 3 days. Afterwards, a hydrophilic polyurethane foam dressing (Kendall, Medtronic, Minneapolis, MN, USA) was applied, which was changed every 5 to 7 days. Eight millimeter punch biopsies (6 mm granulation tissue + 2 mm surrounding skin) were taken 6 and 19 days after the wound creation along the wound edge of the distal limb wounds and fixated in 10% neutral buffered formalin. These two timepoints were chosen because at these moments the acute inflammatory phase and proliferation phase, respectively, are at their maximum in full-thickness wounds on the limbs of horses [18,19].

#### 2.2.2. EGT Wounds

EGT was collected from equine patients treated at the horse clinic for EGT formation in distal limb wounds (Table 1). EGT was resected using a scalpel (Figure 1). These tissue slices were cut into pieces approximately 1 cm long. Immediately upon collection, the tissue samples were fixed in 10% neutral buffered formalin.

### 2.3. Histology

After a fixation of approximately 24 h, the tissues were paraffin embedded using a tissue processor (Thermo Fisher Scientific, STP120 Spin, Waltham, MA, USA) and embedding station (Thermo Fisher Scientific, Microm, EC350, Waltham, MA, USA). Serial sections of 5 µm were cut using a microtome (Thermo Fisher Scientific, HM355S, Waltham, MA, USA) and stretched in a warm water bath (48 °C). Subsequently, the sections were captured on adhesive slides (TrubondTM 380, EMS, 63700-W1, Hatfield, PA, USA) and dried overnight in an incubator at 37 °C.

#### 2.3.1. Histological Staining

Two histological stainings were performed, i.e., the routinely used hematoxylin-eosin staining and the Masson trichrome staining known for its capability to detect collagen fibers. The hematoxylin-eosin staining was performed automatically by the Gemini AS automated slide stainer (Thermo Fisher Scientific, Waltham, MA, USA). For the Masson trichrome staining (Appendix A) the following protocol was performed: after deparaffination and rehydration, the slides were incubated overnight in a Bouin fixative at room temperature to improve the quality of the Masson trichrome staining. After thoroughly rinsing the slides in running tap water, the slides were immersed in Biebrich scarlet-acid fuchsin solution thrice and rinsed in distilled water. Subsequently, the slides were immersed for 20 min in a phosphomolybdic-phosphotungstic acid solution to decolor the collagen followed by an incubation of 8 min in an aniline blue solution. After rinsing in distilled water, the slides were incubated for 8 min in a 1% acetic acid solution. Eventually, the slides were dehydrated using 94% ethanol and isopropyl baths and submerged in a xylene bath for 5 min prior to mounting with DPX (Millipore, Merck, 100579, Darmstadt, Germany).

#### 2.3.2. Immunohistochemical Stainings

On two serial sections an immunohistochemical staining was performed using two different antibodies, i.e., CD163 (clone AM-3K, Abnova, MAB1733, Taipei, Taiwan) and MAC387 (clone MAC387, Thermo Fisher Scientific, MA1-80446, Waltham, MA, USA). Clone AM-3K has been shown to recognize CD163, a commonly used marker for anti-inflammatory macrophages or pro-resolution macrophages [20]. The MAC387 antibody was used to identify cells associated with the primary inflammatory response as MAC387 stains recently recruited monocytes/macrophages and neutrophils. No primary antibody controls were included for every staining (Appendix A). 

The slides were deparaffinated using xylene and hydrated in a decreasing gradient of alcohol. After hydration in distilled water, antigen retrieval was performed using antibody-specific procedures. For the CD163 antibody, the slides were subjected to a heat induced antigen retrieval of 15 min at 110 °C in a decloaking chamber (Biocare Medical, Pacheco, CA, USA) using a target retrieval solution with a pH of 6 (Dako, Agilent, K8005, Santa Clara, CA, USA). For the MAC387 antibody an antigen retrieval with Proteinase K (1/50 in Tris-HCl, Dako, Agilent, S3004, Santa Clara, CA, USA) for 15 min was performed. After the antigen retrieval, the slides were washed with a wash buffer (Dako, Agilent, S3006, Santa Clara, CA, USA) for 3 min before a hydrophobic circle was applied around the sections with a PAP pen (EMS, 71310, Hatfield, PA, USA). The sections were then blocked with 30% rabbit serum. Thereafter, the sections were incubated for 1 h with primary antibodies in a humified chamber at room temperature. The antibodies were diluted in an antibody diluent (Dako, Agilent, S0809, Santa Clara, CA, USA) at the following concentrations: CD163 1/1600 and MAC387 1/2000. After a wash step, the slides were incubated for 5 min in 3% H_2_O_2_ in methanol (Sigma-Aldrich, Merck, 95294-1L, Darmstadt, Germany) to block endogenous peroxidase activity and incubated for 30 min with a 1/200 biotinylated rabbit anti-mouse secondary antibody (Dako, Agilent, E0464, Santa Clara, CA, USA). After washing, an additional incubation with Streptavidin-HRP (1/1000) was performed for 30 min (Dako, Agilent, P0397, Santa Clara, CA, USA). For visualization the slides were incubated with a DAB+ substrate chromogen system (Dako, Agilent, K3468, Santa Clara, CA, USA) for 5 min and subsequently counterstained with hematoxylin for 30 s. After rinsing the slides in tap water for 10 min the slides were dehydrated to xylene and mounted with DPX (Millipore, Merck, 100579, Darmstadt, Germany).

### 2.4. Image Acquisition and Analysis

All slides were photographed and scanned with a 20× objective using an Olympus BX 61 microscope and CellSens software (Olympus, Tokyo, Japan). Subsequently, the pictures were opened and analyzed with QuPath software [21]. First, a grid was applied to every picture with a grid spacing of 500 µm × 500 µm. For each picture four squares were selected randomly: two in a “fibrotic” region and two in an “inflammatory” region. These regions were identified based on the Masson trichrome staining and MAC387 staining, respectively. The selection of these regions was followed by an automatic estimation of the stain vectors for each square. Positive staining was then detected for MAC387 and CD163 using the positive pixel count algorithm with the following parameters: downsample factor: 1; Gaussian sigma: 1 µm; hematoxylin threshold (‘Negative’): 0.1 optical density (OD) units. A DAB threshold (‘Positive’) of 0.08 OD units and 0.03 OD units was applied for the MAC387 and CD163 staining, respectively (Figure 2). 

With the aid of a multiple view window and morphological landmarks, the same regions were analyzed in the serial sections. In the case of an artifact or a tear in the tissue, these were omitted from the selection. That is why for every square not only the number of positive pixels was counted but also the exact surface of the selected area. When analyzing the results, the positive pixels counts were divided through this number and subsequently multiplied by 250,000 (=500 × 500) to take the differences in the analyzed surface area into account. For each tissue block the geometric mean was calculated from the two fibrotic regions and the two inflammation regions. Subsequently, these means were log transformed. 

### 2.5. Statistics

A statistical analysis was performed in RStudio (version 3.6.2) with the aid of the ggpubr (version 0.4.0, https://CRAN.R-project.org/package=ggpubr) and ggplot2 (version 3.3.2, https://CRAN.R-project.org/package=ggplot2) packages, which allowed data visualization [22]. Due to the limited number of horses included in the experimental study no paired statistics could be performed. An unpaired Kruskal–Wallis test was performed to detect the differences between the groups, i.e., EGT wounds, experimental wounds 6 days post-injury and experimental wounds 19 days post-injury. A pairwise comparison was then made between the different groups with the aid of an unpaired Wilcoxon test. A *p*-value of <0.05 was considered significant.

## 3. Results

### 3.1. MAC387 Staining

The MAC387 staining showed a very clear spatiotemporal organization. The majority of the MAC387+ cells were localized near the wound surface (Figure 3). As MAC387+ cells are associated with primary inflammation, i.e., neutrophils, monocytes and macrophages, this staining was used to identify the inflammatory regions. 

In the inflammatory regions, the number of MAC387+ cells in EGT sections resembled most experimental wounds 6 days post-injury (Figure 4B,C). A significant difference (*p* = 0.014) was observed between EGT-developing wounds and physiological healing wounds 19 days post-injury in both the inflammatory and fibrotic regions of the wound (Figure 4). When performing the analysis with only horses in the age range of 7–15 years (age range of the experimental horses), the differences between EGT and d19 wounds became more pronounced in the fibrotic regions (*p* = 0.0043) but less in the inflammatory regions (*p* = 0.052) (Figure 4B,C).

### 3.2. CD163 Staining

CD163+ cells were more uniformly distributed in both the experimental wounds as EGT compared with the MAC387+ cells (Figure 3). No significant difference could be detected regarding the presence of CD163+ cells between the inflammatory and fibrotic regions (Appendix A). In the experimental wounds, the amount of CD163+ cells declined significantly (*p* = 0.0079) between day 6 and day 19 post-injury in the fibrotic/deeper regions of the wound (Figure 5). In the inflammatory or more superficial regions a high variation in the number of CD163+ cells was observed at day 19 post-injury. The amount of CD163+ cells in EGT was similar to the amount observed in the experimental wounds 6 days post-injury (Figure 5). In the fibrotic regions, a significant decrease (*p* = 0.0094) in CD163 staining was seen between EGT wounds and experimental wounds 19 days post-injury (Figure 5A,C). However, when only horses between 7 and 15 years of age were included in the analysis, this difference was less pronounced and not significant any more (*p* = 0.13). 

## 4. Discussion

One of the main characteristics of EGT is the persistent inflammatory response in these wounds. This chronic inflammation may be one of the driving factors stimulating the formation and maintenance of EGT [8]. In physiological healing wounds, the inflammatory phases of wound healing can be subdivided into an onset phase and a resolution phase [23]. In the onset phase, phagocytic neutrophils and M1 macrophages play a crucial role by clearing the wound bed from debris and pathogens. After performing their duty, the neutrophils undergo apoptosis and are ingested by the macrophages. This action, known as efferocytosis, triggers the pro-inflammatory M1 macrophages to switch to anti-inflammatory M2 macrophages in order to resolve the inflammation and initiate the resolution phase of inflammation, which is crucial for the proliferation and remodeling of the healing tissue [16,23]. Here, we used immunohistochemistry to identify neutrophils and macrophages in both the inflammatory/superficial and fibrotic/deeper regions of physiological wound healing by second intention and EGT-developing wounds. 

The inflammatory regions located close to the wound surface contained a lot of MAC387+ cells. The MAC387 antibody is a calprotectin-specific antibody that is expressed on granulocytes such as neutrophils, infiltrating monocytes and potential M1-like macrophages [24]. This MAC387 staining was used to monitor the onset phase of inflammation. In contrast to the MAC387 staining, the CD163+ cells were more uniformly distributed among the wound tissue. These cells represent M2-like macrophages, which are not involved in the onset phase of inflammation. Hence, in contrast to MAC387+ cells, these cells are not predominantly present close to the wound surface. 

In the inflammatory regions, the number of MAC387+ cells seemed to decline between day 6 and 19 after injury in physiological healing experimental wounds. However, in EGT the overall amount of MAC387+ cells remained high, indicative of a persistent inflammatory response in EGT. As the wound bed was still open in both the experimental and EGT wounds, it is not surprising that primary inflammatory cells remained present in the upper parts of the wound. In the fibrotic regions of the wound, the MAC387 staining was significant higher in EGT-developing wounds compared with the physiological healing wounds. This in indicative of a persistent inflammatory response in EGT-developing wounds.

The CD163 staining in the inflammatory regions showed a high variability. This can probably be explained by the fact that this region was located near the wound surface and is strongly influenced by external factors including the degree of contamination and the amount of wound exudate.

A difference was also observed in CD163 staining between the wound types in the fibrotic regions of the wound. Between day 6 and day 19 after injury, a significant drop in CD163+ cells was observed in the fibrotic regions of physiological healing wounds. CD163+ staining in EGT was more variable. Higher levels of CD163 cells were found in EGT wounds compared with the physiological healing wounds 19 days after injury but when the analysis was only performed on the six horses with ages in the same interval as the horses from the experimental study, the differences became less pronounced. This may indicate an age effect but the number of horses in each age interval was too small to verify this.

In physiological wound healing, the number of M2-like macrophages declines when the wound transits from the proliferation to the remodeling phase. As the number of M2-like macrophages in EGT resembles more closely the physiological wounds of 6 days compared with the physiological wounds of 19 days, this may suggest that EGT consists of young fibrotic tissue stuck in the proliferation phase, which is unable to transit to the remodeling phase. 

This may also explain the excessive fibroplasia typically seen in EGT. Excessive M2 macrophage activation has been associated with scarring and fibrosis [25,26]. This may be due to the production of transforming growth factor beta (TGF-β). M2 macrophages are known to regulate fibrosis via the production of TGF-β and it has been shown that horse limb wounds are characterized by persistently high levels of TGF-β [27]. Aside from regulating fibrosis, TGF-β is also known to be involved in angiogenesis, inflammation, collagen synthesis and extracellular matrix deposition and remodeling [28].

Human diabetic ulcers are characterized by a chronic but low grade inflammation similar to EGT wounds [5,29,30,31]. In contrast to EGT wounds, the formation of granulation tissue is very limited and the wound remains open, which can even lead to amputation [29]. Interestingly, non-healing diabetic foot ulcers are associated with a failed M2 macrophage development [31,32,33]. It is tempting to speculate that EGT wounds suffer from both a chronic onset and a chronic resolution phase of inflammation due to high levels of primary immune cells and M2 macrophages whereas diabetic foot ulcers in humans suffer from a chronic inflammation caused mainly by M1 macrophages. This could explain the phenotypic differences, i.e., the hyperproliferation of the granulation tissue versus a deficit granulation formation, respectively. Keloids, a fibroproliferative disorder seen in humans, are also associated with excessive amounts of M2 macrophages [34]. All these types of impaired wound healing do not succeed to complete the wound healing process as they seem to be arrested along the way. 

It is of note to mention that the distinction between M1 and M2 macrophages is not strict and that these two types of macrophages need to be considered as the two extremes of a whole spectrum [35]. For example, based on in vitro experiments, M2 macrophages can be divided in several subtypes, namely, M2a, M2b, M2c and M2d, according to the inducing agent and the expressed markers [36]. It remains unclear if all these subtypes are expressed in vivo [36]. The expression of the CD163 marker also differs between these subsets [36]. From M2a and M2c macrophages it is known that they express CD163 [37]. It is challenging to identify different macrophage subsets due to a lack of unique markers. Moreover, markers are rarely validated for horse tissue, which makes it very cumbersome to optimize immunohistochemical stainings for equine tissue. In the future, more research is needed on the different macrophage subtypes in horses. RNA studies on single cell levels can help to identify the different cells and their subtypes that are important during the different phases of wound healing. In a study performed by Theoret et al. (2013) only a limited number of macrophages were detected in EGT [38]. This discrepancy can possibly be explained by the fact that a different marker was used, i.e., anti-CD68. CD68 is expressed by cells from the monocyte lineage and has been frequently used as pan-macrophage marker.

One of the limitations of this study is the limited number of horses included in the study. Another limitation was the fact that EGT used in this preliminary study originated from equine patients with a variable age, wound history and wound age. Performing a follow-up study on a much larger patient cohort would make it possible to investigate if these parameters have an effect on EGT. It would also be interesting to sample EGT wounds at different time points to see how EGT develops over time. 

## 5. Conclusions

Our results showed a similar abundant number of macrophages in the early phase of physiological healing (6 days post-injury) and EGT, stressing the importance of these cells in equine wound healing and EGT formation. EGT could be very useful in future research to study the inflammatory phases and the role of different pro-fibrotic (M2) macrophages (sub)types in wound healing.

## Figures and Tables

**Figure 1 animals-11-02728-f001:**
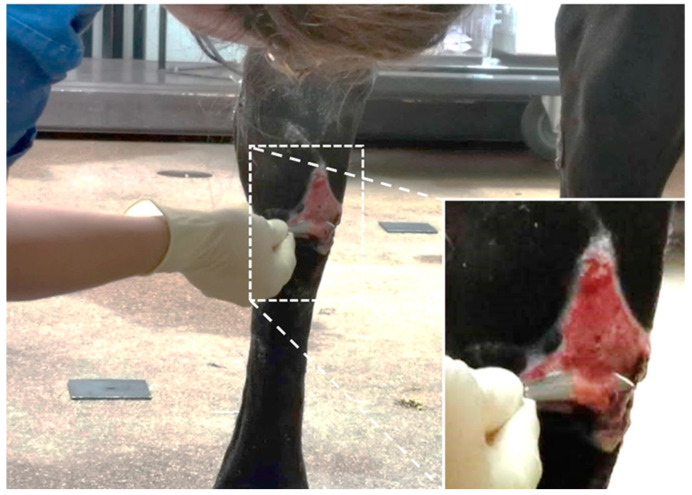
Picture of an 8-year-old mare, which came to the clinic with a hypergranulating wound at the distal aspect of the right tarsus. Exuberant granulation tissue is trimmed with a scalpel.

**Figure 2 animals-11-02728-f002:**
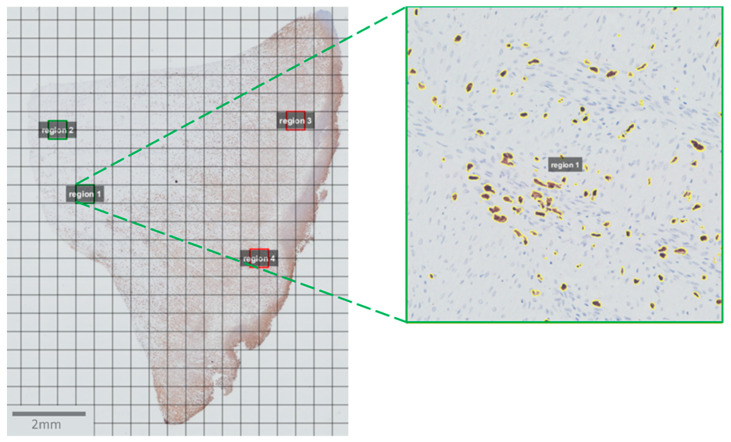
Example of the analysis of the tissue slices in the QuPath software. After importing the scanned images, a grid was placed on the pictures. For each tissue section, four squares of 500 µm × 500 µm were selected: two squares in a fibrotic region (region 1 and 2, green) and two squares in an inflammation region (region 3 and 4, red). On each of these four squares the positive pixel count algorithm was run. Pixels that were counted as positive pixels are encircled in yellow (insert).

**Figure 3 animals-11-02728-f003:**
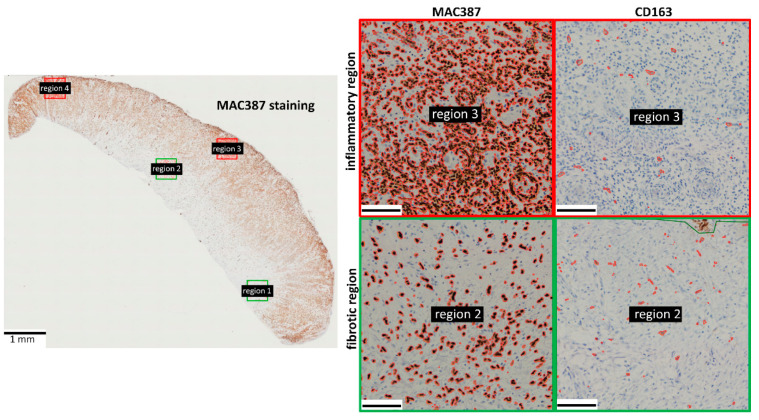
Immunohistochemistry showing the spatial organization of MAC387+ cells and CD163+ cells in horse EGT. The majority of the MAC387 staining was found in the inflammatory regions (indicated in red) near the wound surface. The CD163 staining was more uniformly distributed. Stained pixels were detected using the positive pixel count algorithm in QuPath and encircled in red. Scale bar inserts = 100 µm. Red frame: inflammatory regions; green frame: fibrotic regions.

**Figure 4 animals-11-02728-f004:**
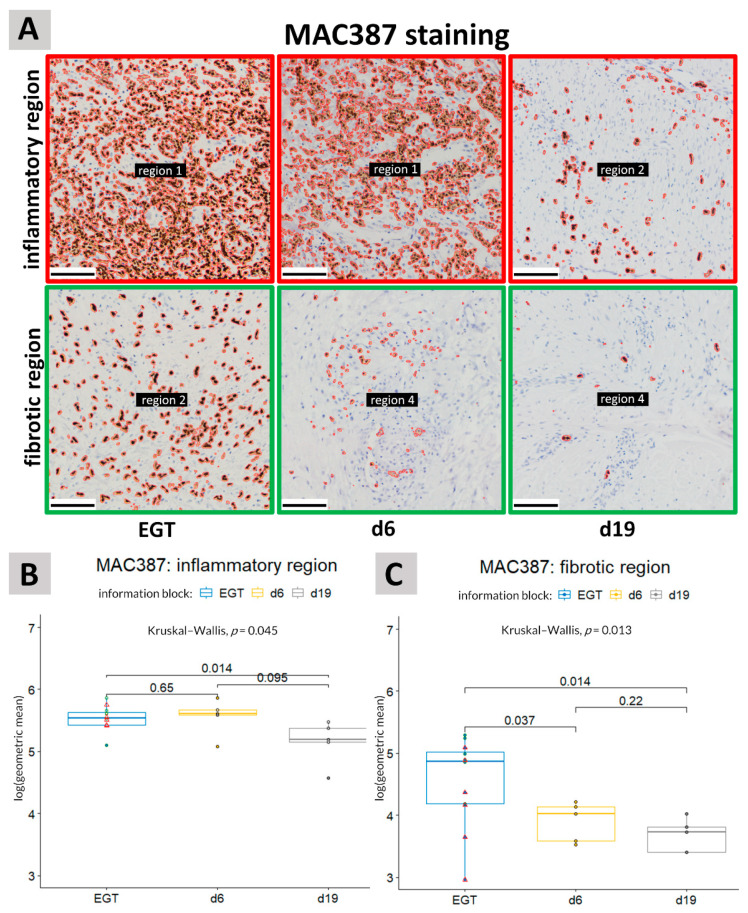
MAC387 staining. (**A**): Representative pictures of immunohistochemical stainings showing the differences in MAC387 immunolabeling between the inflammatory and fibrotic regions in EGT and physiological healing in experimental wounds at 6 days (d6) or 19 days (d19) post-injury. Scale bar = 100 µm. (**B**,**C**): MAC387 staining was quantified in sections of horse wound tissue in the inflammatory regions (**B**) and in the fibrotic regions (**C**). An unpaired Kruskal–Wallis was performed to detect the differences between the different groups, i.e., EGT wounds, d6 and d19. A pairwise comparison was then made between the different groups with the aid of an unpaired Wilcoxon test. A significant difference was detected between EGT-developing wounds and physiological healing wounds of 19 days in both the fibrotic and inflammatory regions of the wound. A *p*-value of <0.05 was considered significant. Green circles represent horses with an age between 7 and 15 years (age range of the experimental horses) and red triangles represent younger horses. See also Appendix A for the unprocessed pictures.

**Figure 5 animals-11-02728-f005:**
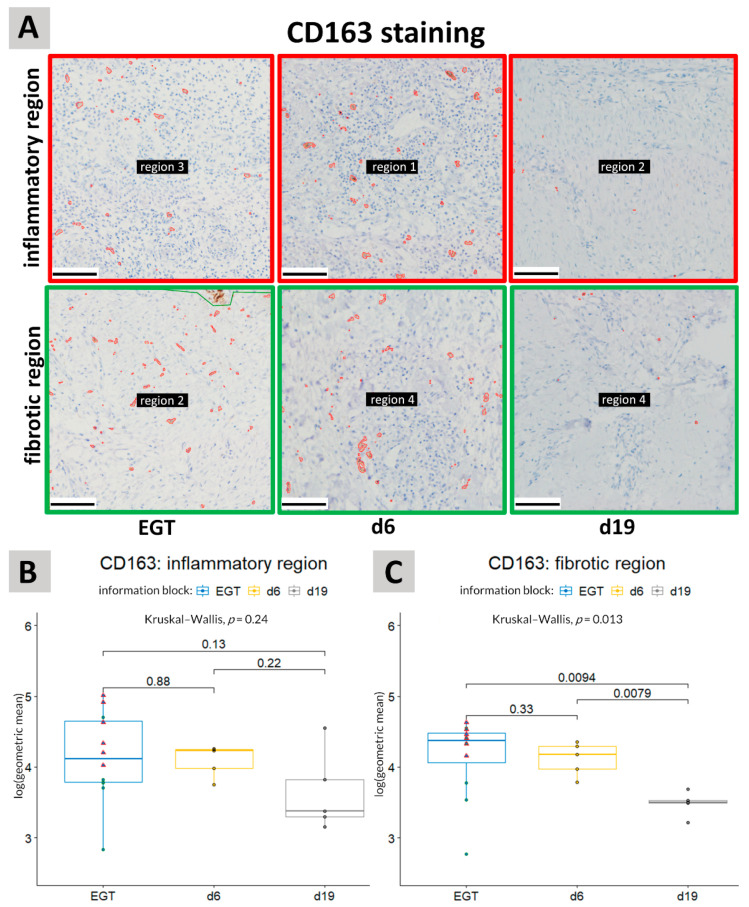
CD163 staining. (**A**): Representative pictures of immunohistochemical stainings to show the differences in CD163 immunolabeling between the inflammatory and fibrotic regions in EGT and physiological healing in experimental wounds at 6 days (d6) or 19 days (d19) post-injury. Scale bar = 100 µm. (**B**,**C**): CD163 staining was quantified in sections of horse wound tissue in the inflammatory regions (**B**) and in the fibrotic regions (**C**). An unpaired Kruskal–Wallis was performed to detect the differences between the different groups, i.e., EGT wounds, d6 and d19. A pairwise comparison was then made between the different groups with the aid of an unpaired Wilcoxon test. No significant differences were detected between the groups in the inflammatory regions. In the fibrotic regions, a significant difference was observed between wounds 19 days post-injury and both EGT and wounds 6 days post-injury. A *p*-value of <0.05 was considered significant. Green circles represent horses with an age between 7 and 15 years (age range of the experimental horses) and the red triangles represent younger horses. See also Appendix A for the unprocessed pictures.

**Table 1 animals-11-02728-t001:** Table with additional information on the exuberant granulation tissue-developing wounds. RH: right hindlimb; LH: left hindlimb; RF: right front limb; LF: left front limb.

Horse	Wound Age (Days)	Affected Limb	Localization EGT	Horse Age	Horse Weight (kg)
1	23	RH	Distal dorsal tarsus	~8 years	600
2	26	LH	Distal plantar tarsus	~2 years	405
3	16	RH	Distal dorsal tarsus	~6 years	610
4	26	LH	Metatarsus both lateral and medial	~2 months	/
5	Presented at the clinic with an old wound, exact age of wound was unknown	RH	Medial tibia	~1 year	340
6	30	RF	Carpus	~11 years	635
7	Presented at the clinic with an old wound, exact age of wound was unknown	RH	Tarsus + metatarsus	~11 years	593
8	66	LF	Fetlock	~14 years	480
9	17	LH	Fetlock	~11 years	/
10	Presented at the clinic with an old wound, exact age of wound was unknown	RF	Metacarpus	~9 years	/
11	9	LH	Metacarpus	~5 years	520
12	20	RH	Tarsus	~5 years	465

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
