# Peer review of "High Numbers of CD163-Positive Macrophages in the Fibrotic Region of Exuberant Granulation Tissue in Horses"

_animals, 2021, doi:10.3390/ani11092728_

Round 1
Reviewer 1 Report
Please see attached document.

Reviewer 2 Report
The manuscript animals-1288089 describes cases of CD163-positive macrophages in the fibrotic region of exuberant granulation tissue in horses.
The report is interesting and current, however, the manuscript deserves some major revision before being reconsidered for publication.
In my opinion, the authors should not refer to the M1 and M2 macrophages, the results are not supported by the primary antibodies specific for the M1 and M2 macrophages.
In the evaluation of the positivity of MAC387 macrophages, one can only hypothesize a positivity to M1 but in the discussions.
I suggest excluding the superficial inflammatory areas and showing only the areas of fibrosis.
In the experiment, the horses underwent various treatments.
Why didn't they pay attention to the differences in results? Why has no correlation been made between treatment and macrophages?
What criteria used for including and excluding animals during the experiment?
ABSTRACT
Line 24: change significant with “significantly”
Line 27: change to a plural “processes”
Line 28: In the keywords section, it is recommended to delete M1 and M2 .
INTRODUCTION
Line 34: change “this” with “these”
Line 40 to 43: include image and caption to illustration of “fig.1” in materials and methods
Line 61: the phrase “over the course of the wound healing” may be redundant ….” throughout the wound healing”
MATERIALS AND METHODS
Line 79: in the phrase “from of a previous” delete "of"
Line 100: change limbs with “limb”
Line 121: “were fixed in 3.5% formaldehyde” ???
Line 138: adjust the verbe “recognize” with “recognizes”
Line 140 to 144: these speculations do not insert into materials and methods ... eventually transfer into discussions
Line 145: delete “a”
Line 145: why “a no primary antibody control was included” ? If it is possible, I would like to see the negative control because the chromogen DAB often stains inflammatory cells unspecifically.
Line 159: conjugated rabbit anti-mouse….??
Line 169: change “his” with “its”
Line 189: change “this” with “these”
RESULTS
Line 222: change “was” with “were”
Line 241: delete “as”
Line 265: delete “as”
DISCUSSION
Line 295: change “for” with “of”
Line 296: change surprisingly with “surprising”
Line 300: “is located” instead “in located”
Line 313: persistently instead persistent
Line 323: change “human suffer” with “humans suffering”
Line 333: “these subsets” instead “this subset”
Line 341: Change “In a study of “with “In a study by”
Reviewer 3 Report
I think that the study, despite the low number of subjects, shows some meaning results, but I also think that the authors should be more critical underling the limits of the study, such as, to me, the low number of investigated cases but also the small numbers of sampling of the EGT lesions. Experimental lesions show a temporal evolution of the inflammatory process, while EGT lesions are examined just once, and there is no observation of an eventual evolution of the inflammatory process that could be different in the composition of the cells involved compared to experimental healing wounds, as supposed by authors, but also that could be hypothetically characterized by diverse duration of the early and late phases.
There is, also, no mention to the treatment used for EGT before sampling. To compare the lesions, the groups should be uniform on this point of view. Do we know, indeed, if the the treatment influenced the results?
Line 10 write “(EGT)” instead of “or EGT”
Line 11: delete the comma after the word but.
Line 11: I would write “is supposed to be an importante contributing factor” instead of “probably an important contributing factor”.
Line 15: write “higher” instead of “high”
Line 16 I would write “This suggests that EGT wounds could not be able to proceed through further phases in the wound healing process” and I will add “or that the early phase of inflammation is prolonged”
Line 26: you say that the persistent high amount of fibrosis promotes CD163+ cells, but later at lines 66-68 you say that the M2-type macrophages orchestrate the proliferation phase of wound healing and are involved in the formation of granulation tissue. The 2 sentences are contradictory, please, explain and adjust accordingly.
Line 40: delete ‘(Figure 1)’ and move it to line 119 after ‘EGT was resected with the aid of scalpel.’
Line 79: I would write “derived from” instead of “originated”
Line 100 write “one limb” instead of “one limbs”
Line 134: I would move here the Histological staining section (lines 167-181, as usually the histology is perform before immunohistochemistry, to be sure to have representative slides of the lesion.
Lines 137-144: this part does not belong to the material and methods section, move it to the results “Clone AM-3K, was used to identify M2-like macrophages as it has been shown that it recognize CD163, a commonly used marker for anti- inflammatory macrophages or pro-resolution macrophages [18]. The MAC387 antibody was used to identify cells associated with the primary inflammatory response as MAC387 stains recently recruited monocytes/macrophages (potentially M1-like macrophages). But it is important to note that this calprotectin specific antibody will also detect granulocytes such as neutrophils. As neutrophils are the first cells to arrive at the site of injury, we can state that MAC387+ cells are associated with the primary inflammatory response.“
Line 168: probably, I would write routine Hematoxylin-Eosin instead of general.
Line 171-172: Write, please: For the Masson (Figure S1)trichrome staining the following protocol was performed: after deparaffination and rehydration, the …etc.
Line 187: write ‘(Figure 2)’ instead of ’(Figure 3)’
Line 193: remove ‘(Figure 2)’
Lines 219-220: The results of the histology are completely missed! Please, add careful descriptions of the lesions and the results of the different stains pointing at the differences of the pattern of the 3 groups of lesions. There was disepitelization in all 3 groups? This is important for your conclusions (see, for example, lines 297-301)
Lines 227-228: write “Figures 4B, 4C” instead of “Figure 4 B&C”
Lines 227-228: “EGT sections resembled most experimental 227 wounds 6 days post injury (Figure 4 B&C).“ Please, before announcing this conclusion, describe carefully the microscopic pattern observed.
Lines 230-233: At the beginning of the legend add also the organ, age of the wound, specify that this is the experimental wound, and add the technique (as: Skin, 6 day old experimental wound, horse. Immunohistochemistry for MAC387 and CD163.
Write, please, “immunolabeled cells” instead of staining at line 230. Add the meaning of the green frame. Furthermore, usually in the figures the skin is oriented with the surface/epidermis on the top. Can you please rotate your left image following this conventional scheme?
Line 235: can you add a description for panel A, please?
Line 236: write ‘6 days (d6) or 19 days (d19) post injury’
Line 242: remove d6: experimental wounds 6 days post injury. d19: experimental wounds 19 days post injury.
Lines 245-246: what kind of wounds? I think you mean experimental wound, please adjust.
Lines 251-252: The sentence “The staining of CD163 in EGT was very similar to the one observed in experimental wounds 6 days post injury” is not a description of what you saw, please describe before the pattern found.
Line 254: write “Figures 5A, 5C ” instead of “Figure 5A & C ”
Figure 5: the name of the region in the center of each image is not readable, too small
Line 257: Can you add a description for panel A, please?
Line 258: write ‘6 days (d6) or 19 days (d19) post injury’
Line 266: remove d6: experimental wounds 6 days post injury. d19: experimental wounds 19 days post injury.
Lines 283-291: please, rephrase the entire period. There are too many repetition of the mechanism and inflammatory phases, already described in lines 269-282. You can instead describe what you found and what was expected and was not expected in your groups of samples.
Lines 293-294: write, please, the number of MAC387+ cells seemed to decline between day 6 and 19 after injury in the experimental wounds.
Lines 341-343. If I have correctly understand what you want to say, I believe that these sentences are related to the previous period and should then be attached to it at line 340.
Lines 346-347: write “healing wounds in horses” instead of “healing horse wounds”
Lines 346-350 First of all, I have a question: how can a pathological entity, as the EGT; be a model for physiological healing? I wouldn’t propose it as a model.
I would change the conclusions are or less as follow:
Our results show a similar abundant amount of macrophages in early phase of physiological healing (6 days post injury) wounds and in late phase of EGT, stressing the importance of these cells in equine wound healing process and EGT formation. EGT could then be useful in future research to study the inflammatory phases and the role of different pro-fibrotic (M2) macrophages (sub)types in wound healing.
Supplemental files: please, follow the instructions to authors and name files with the same style, not Figure S1 and Figures 1S in the same text.
Reviewer 4 Report
Comments to the Authors
The article described the profile of macrophage phenotypes in exuberant granulation tissue (EGT) in comparison to the changes in macrophage phenotypes in surgical wounds healing by secondary intention in horses. The authors have revealed that the population of proinflammatory macrophages was generally constant during the healing of experimental wounds and similar in EGT but the population of pro-resolution (M2) macrophages decreased with time in fibrotic regions of surgical wound but not in EGT.
The study is well designed but my major concern is the low number of macrophages (as reported previously by Theoret et al. 2013 and other authors) in EGT, as visible on raw figures. That is why I would expect more information regarding:
- the composition of EGT (and granulation tissue)
- the conditions that support EGT formation
- the role of M2 macrophages (cytokine signaling) in healing by secondary intention and in EGT
I suggest to add these issues in the introduction. It will help in better understanding of M2 decrease in surgical wounds vs. persistence in EGT. I would expect more regarding modulatory effects of M2 in the discussion.
My second concern is the lack of the limitations of the study. There is nothing wrong in limitations, adding such section can even help in better understanding the results.
Specific comments:
Line 86 – 2 months old foal is quite different from adult 8 years old horse. Was there a difference in the composition of EGT in the foal and adult horses? How about the horse with old wound?
Line 225 – “There was a trend” – the results were not significant, so there was no difference and it should not be interpreted as a trend
Lines 320-326: diabetic foot ulcers do not seem good for comparison, as they do not occur in horses and are not similar to any equine disease. I think it is better to compare human keloid and EGT
Round 2
Reviewer 2 Report
Authors followed all the suggestions given. Define the study as a "preliminary study". In fact, the deficiencies observed could be premise for subsequent experiments. In my opinion, stated all the suggestions accepted and the corrections done, the manuscript has been improved and is now suitable for publication
Reviewer 3 Report
Dear Authors,
thank you for your work, that, together with the suggestions of the other reviewers, improved a lot. This is why I think your work is suitable for publication.
Please, adjust only these few things:
Figure 1: spell out EGT
Line 138-139 the line is repeated twice
Please, spell out EGT in the table title
Lines 239-241: the lines are repeated 3 times.
Probably is a problem of the pdf file on my computer, but I see figure 5 as a replication of figure 4.
